# HAD-Net: A Hierarchical Adversarial Knowledge Distillation Network for Improved Enhanced Tumour Segmentation Without Post-Contrast Images

**Saverio Vadacchino**[1]                                    SAVERIO@CIM.MCGILL.CA
**Raghav Mehta**[1]                                          RAGHAV@CIM.MCGILL.CA
**Nazanin Mohammadi Sepahvand**[1]                           NAZSEPAH@CIM.MCGILL.CA
**Brennan Nichyporuk**[1]                                    BRENNANN@CIM.MCGILL.CA
**James J Clark**[1]                                         CLARK@CIM.MCGILL.CA
**Tal Arbel**[1]                                             ARBEL@CIM.MCGILL.CA

[1] *Centre for Intelligent Machines (CIM), Department of Electrical and Computer Engineering, McGill University, Montreal, Quebec, Canada.*

## Abstract

Segmentation of enhancing tumours or lesions from MRI is important for detecting new disease activity in many clinical contexts. However, accurate segmentation requires the inclusion of medical images (e.g., T1 post-contrast MRI) acquired after injecting patients with a contrast agent (e.g., Gadolinium), a process no longer thought to be safe. Although a number of modality-agnostic segmentation networks have been developed over the past few years, they have been met with limited success in the context of enhancing pathology segmentation. In this work, we present HAD-Net, a novel offline adversarial knowledge distillation (KD) technique, whereby a pre-trained teacher segmentation network, with access to all MRI sequences, teaches a student network, via hierarchical adversarial training, to better overcome the large domain shift presented when crucial images are absent during inference. In particular, we apply HAD-Net to the challenging task of enhancing tumour segmentation when access to post-contrast imaging is not available. The proposed network is trained and tested on the BraTS 2019 brain tumour segmentation challenge dataset, where it achieves performance improvements in the ranges of 16% - 26% over (a) recent modality-agnostic segmentation methods (*U-HeMIS*, *U-HVED*), (b) *KD-Net* adapted to this problem, (c) the pre-trained student network and (d) a non-hierarchical version of the network (*AD-Net*), in terms of Dice scores for enhancing tumour (ET). The network also shows improvements in tumour core (TC) Dice scores. Finally, the network outperforms both the baseline student network and *AD-Net* in terms of uncertainty quantification for enhancing tumour segmentation based on the BraTS 2019 uncertainty challenge metrics. Our code is publicly available at: https://github.com/SaverioVad/HAD_Net

**Keywords:** Knowledge Distillation, Adversarial, Discriminator, Hierarchical, Enhancing tumour, Missing Sequence, Contrast Enhancement

## 1. Introduction

The inclusion of different MRI sequences (e.g. T1, T2, FLAIR) (Van Tulder and de Bruijne, 2015; Bakas et al., 2018) greatly improves the performance of automatic tumour or lesion segmentation from magnetic resonance imaging (MRI). In particular, the presence of contrast enhanced T1-weighted (T1ce) MRI has been shown to play a crucial role in

automatic segmentation of enhancing tumours or lesions from MRI, and therefore is important for determining treatment efficacy, patient prognosis (Pallud et al., 2009) and tumour grading (Upadhyay and Waldman, 2011). However, acquiring T1ce images involves injecting a patient with a contrast agent (e.g. Gadolinium), a process that is invasive and no longer thought to be safe for patients (Perazella, 2008). Although challenging, an automatic segmentation technique that can accurately segment enhancing tumour or lesions reliably without requiring T1ce images would have an enormous impact on patient care.

Recently, a few deep learning techniques have been introduced to address the problem of tumour segmentation with missing MRI sequences, however none have specifically focused on missing T1ce. This includes networks designed to synthesize the missing MRI sequences using a Generative Adversarial Network (GAN) (Sharma and Hamarneh, 2019) or a Convolutional Neural Network (CNN) (Mehta and Arbel, 2018), and then use them to improve the downstream segmentations (Van Tulder and de Bruijne, 2015). Several modality-invariant techniques, such as *U-HeMIS* (Havaei et al., 2016) and *U-HVED* (Dorent et al., 2019), were developed to segment tumour sub-tissues given any combination of available MR sequences. However, the performance of these models in segmenting enhancing tumours degrades substantially when T1ce is missing. Additionally, a recent knowledge distillation (KD) network *KD-Net* (Hu et al., 2020) has shown some success in segmenting brain tumours when only one MR sequence (e.g., T1ce) is provided during inference.

In this paper, we introduce *HAD-Net*, a novel hierarchical adversarial KD network, where a pre-trained teacher, with access to all images, is used to teach a student network to better perform segmentation when crucial information, here T1ce, is absent during inference. This is achieved through a hierarchical discriminator which distills the latent information encoded in the teacher's feature maps to the student at different resolution scales. While hierarchical discriminators exist, they can only be found in the generative modeling literature (e.g. GAN), without KD (Karnewar and Wang, 2019; Valvano et al., 2020). Furthermore, while adversarial KD techniques already exist, they are not hierarchical in nature (Zhang et al., 2020; Chung et al., 2020). Therefore, to the best of our knowledge HAD-Net is the first method to effectively combine these two components to permit KD to better overcome the large domain shift arising from the absence of crucial information during inference.

We evaluate our method on the Brain Tumour Segmentation (BraTS) 2019 challenge dataset (Menze et al., 2014), where *HAD-Net* achieves performance improvements over modality-agnostic segmentation methods such as *U-HeMIS* and *U-HVED* in terms of Dice scores for enhancing tumour (ET), by 18.9% and 26.0% respectively, as well as improvements of 17.2% over *KD-Net+*, a variant of *KD-Net* adapted to this context (see section 3.1). Our network also shows improvements in ET Dice scores over both the baseline student network (without the teacher) and a non-hierarchical version of the network (*AD-Net*), by 16.2%, and 18.7% respectively. In addition, *HAD-Net* also shows some relatively smaller improvements in tumour core (TC) Dice scores. Finally, *HAD-Net* shows improvements in quantifying uncertainty in the resulting ET segmentations over the pre-trained student network and *AD-Net*, based on the metric from the BraTS 2019 uncertainty quantification challenge (Mehta et al., 2020).

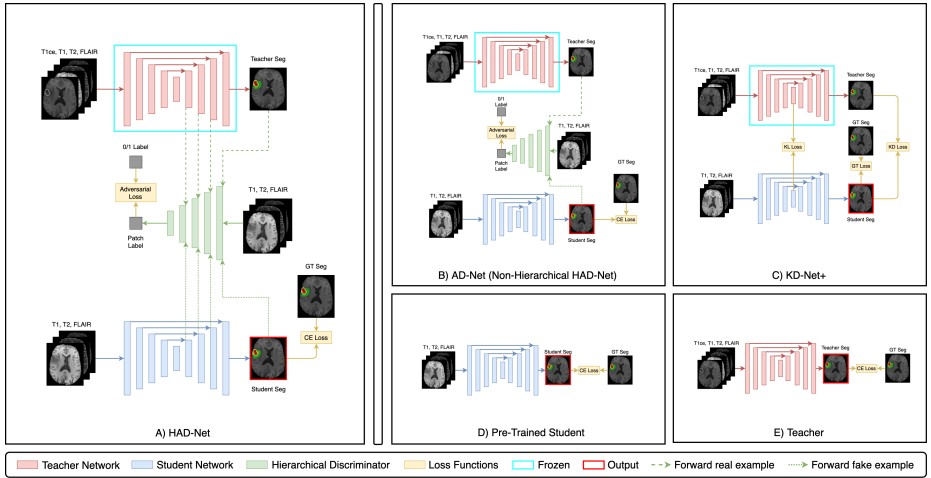

**Figure 1:** Diagram of *HAD-Net* (A), and several competing variants (B-E). Note that for *HAD-Net*, *AD-Net*, and *KD-Net+*, the teacher network is frozen during training, and only the student network is used during inference.

## 2. HAD-Net

### 2.1. Method

The HAD-Net (Hierarchical Adversarial Distillation Network) architecture consists of three main components: (1) the teacher network, (2) the student network, and (3) the hierarchical discriminator (HD). A diagram of the architecture is shown in Figure 1(A).

In this work, we focus on multi-class tumour segmentation. In this context, the teacher network has access to all available MRI sequences as input (e.g., T1ce, T1, T2, and FLAIR), while the student network only has access to the pre-contrast MRI sequences, $\mathbf{X}_{pre}$ (e.g. T1, T2, FLAIR). The HD component attempts to bridge the domain gap between the student and the teacher by mapping their segmentations ($\mathbf{S}_{seg}$ and $\mathbf{T}_{seg}$) as well as their multi-scale feature maps ($\mathbf{S}_{latents}$ and $\mathbf{T}_{latents}$) to a common space. This is done by concatenating the HD's features with the corresponding multi-scale feature maps from either the student or the teacher network, which is demonstrated in Figure 1(A). This is in contrast to the more common, non-hierarchical adversarial distillation network, henceforth denoted *AD-Net*, which only provides the student and teacher final segmentations as inputs to the Discriminator (see Figure 1(B)). By forwarding the hierarchical latent representations to the HD, global and local information is distilled. Furthermore, a pathway is created between the discriminator and the student network that facilitates gradient flow. This helps to address the issue of vanishing gradients, a problem which plagues many modern adversarial networks (Wiatrak et al., 2019; Barnett, 2018).

The role of the HD is to try to distinguish the teacher's segmentations and intermediate latent representations from the student's. In classical terminology popular in the GAN literature (Goodfellow et al., 2014), the student acts as the generator, generating "fake" data, the teacher's segmentations and intermediate latent representations act as the "real" data, and the HD attempts to distinguish between the "real" and "fake" data samples.

Through the classical adversarial game, the domain shift is addressed as the student learns to bring its segmentation and underlying latent representations closer to those of the teacher.

Prior to training *HAD-Net*, both the student (Figure 1(D)) and the teacher (Figure 1(E)) networks are pre-trained on the task of multi-class tumour segmentations [1]. During the training of *HAD-Net*, the teacher network is frozen, while the weights of the student network and the HD are updated. During inference, the student performs multi-class tumour segmentations without contrast-enhanced images (eg. T1ce). Although there will likely be some loss of accuracy when compared to the segmentation results of the teacher, it is anticipated that learning from the teacher will lead to significant improvements over baseline methods.

### 2.2. Network Architecture

The student and the teacher networks are 3D U-nets (Çiçek et al., 2016) adapted from the No New-Net model (Isensee et al., 2018). The Hierarchical Discriminator (HD) is a fully convolutional hierarchical patch-based discriminator (Isola et al., 2017; Cirillo et al., 2020)[2].

### 2.3. Loss

The student network is trained using the student loss, denoted as $L_S$, which consists of the weighted combination of two terms: (1) a weighted cross-entropy (CE) loss term between the student network segmentation, $\mathbf{S}_{seg}$, and the ground truth segmentation, $\hat{y}$, and (2) a Mean Squared Error (MSE) adversarial loss term (Mao et al., 2017)(see Equation 1). The overall $L_S$ loss ensures that the student and teacher network outputs and features are mapped to a common representation, and that this ultimately translates to improved segmentation performance for the student network.

$$L_S = CE[\mathbf{S}_{seg}, \hat{y}] + \lambda * MSE[HD(X_{pre}, \mathbf{S}_{seg}, \mathbf{S}_{latent}), \mathbf{1}] \tag{1}$$

The HD is trained using the LS-GAN (Mao et al., 2017) loss, denoted as $L_{HD}$. It is made up of two Mean Squared Error (MSE) loss terms (see Equation 2). One term is between the HD output, after being passed a "fake" data sample from the student, and a tensor of all zeros (Isola et al., 2017). The other term is the MSE loss between the HD output, after being passed a "real" data sample from the teacher, and a tensor of all ones. The overall $L_{HD}$ loss ensures that the HD is able to properly distinguish the student's output and multi-scale features from those of the teacher, which ultimately allows for the distillation of meaningful information to the student network via the adversarial component of $L_S$.

$$L_{HD} = MSE[HD(X_{pre}, \mathbf{S}_{seg}, \mathbf{S}_{latent}), \mathbf{0}] + MSE[HD(X_{pre}, \mathbf{T}_{seg}, \mathbf{T}_{latent}), \mathbf{1}] \tag{2}$$

## 3. Experiments and Results

### 3.1. Experiments

In this paper, our experiments are focused on brain tumour sub-structure segmentation from MRI without post-contrast T1 images (T1ce). The specific objectives of the experiments

---

1. The details of the pre-training procedure can be found in Appendix B.
2. Additional architectural details pertaining to each component of HAD-Net can be found in Appendix A.

| Model | Teacher | Pre-Trained Student | HAD-Net | AD-Net | KD-Net+ | U-HeMIS | U-HVED |
|-------|---------|---------------------|---------|--------|---------|---------|--------|
| **WT** | $89.6 \pm 08.0$ | $87.9 \pm 12.4$ | $87.5 \pm 07.6$ | $88.1 \pm 09.4$ | $88.9 \pm 08.2$ | $86.9 \pm 11.7$ | $86.9 \pm 11.9$ |
| **TC** | $79.1 \pm 20.7$ | $62.5 \pm 21.9$ | $\mathbf{66.7 \pm 19.7^*}$ | $64.5 \pm 17.2$ | $63.3 \pm 17.2$ | $64.3 \pm 18.3$ | $61.5 \pm 21.0$ |
| **ET** | $73.3 \pm 30.2$ | $34.3 \pm 25.3$ | $\mathbf{39.8 \pm 26.9^{**}}$ | $33.5 \pm 22.2$ | $33.9 \pm 22.9$ | $33.5 \pm 23.3$ | $31.6 \pm 20.9$ |

**Table 1:** Table showcasing the percentage (%) Dice scores (mean ± std) achieved by each method on the BraTS 2019 Validation set for whole tumour (WT - ■■■), tumour core (TC - ■■), and enhancing tumour (ET - ■). (*) and (**) denote a statistically significant difference ($p < 0.05$ and $p < 0.001$) between *HAD-Net* and all other methods, determined using a two-sided paired t-test. Note that only the Teacher network had access to the T1ce sequence, and therefore performed the best. Also, note that all methods (aside from the teacher) performed similarly for WT segmentation, with no statistically significant differences.

are to show improvements in segmenting enhancing tumour, a problem that is yet to show good results and whose clinical implications are important in a number of domains, while maintaining or improving the segmentation results for the other structures. To that end, we compare *HAD-Net* to the modality-agnostic methods, *U-HeMIS* and *U-HVED* [3]. We also compare to *KD-Net+* (Figure 1(C)), a baseline which maintains the original *KD-Net*'s KD framework but, for a fair comparison, replaces its student and teacher networks with HAD-Net's. To evaluate the advantages of a hierarchical discriminator, we compare *HAD-Net* with a non-hierarchical adversarial distillation network, *AD-Net* (Figure 1(B)). Results from the pre-trained student (Figure 1(D)) and teacher networks (Figure 1(E)) act as lower and upper bounds for performance comparisons. In addition to traditional Dice measures, we also examine the performance of the proposed method in the context of uncertainty quantification, specifically exploring if the model is correct when it is confident, and more uncertain when it is incorrect (Mehta et al., 2020) [4].

### 3.2. Dataset

For our experiments, all methods make use of the MICCAI BraTS 2019 challenge training and validation datasets (Menze et al., 2014; Bakas et al., 2018). The training set consists of 335 cases (259 High-Grade Glioma (HGG), and 76 Low-Grade Glioma (LGG) patients), while the validation set consists of 125 cases. Four MRI sequences are available for each patient: T1, T1ce, T2, and FLAIR. The challenge provides labels for *only* the training dataset in the form of segmentation maps depicting 3 tumour sub-tissues: necrotic non-enhancing tumour core (■), peritumoral edema (■), and GD-enhancing tumour (■) , and the background (■). In order to provide a qualitative analysis and comparison of the methods, the training set was randomly split into a training, validation, and testing set (200:66:69 split). The HGG and LGG samples were split proportionally between these sets[5].

We evaluated the quantitative performance of *HAD-Net* and the competing methods by uploading the segmentation maps produced on the BraTS 2019 validation set. Quantitative assessment was provided by the organizers, thus permitting objective assessment of

---

3. We used the publicly available code for *U-HeMIS* and *U-HVED* provided by the authors: https://github.com/ReubenDo/U-HVED.

4. See appendix D for more details on the implementation of the competing methods.

5. More information outlining the data pre-processing procedure can be found in Appendix C.

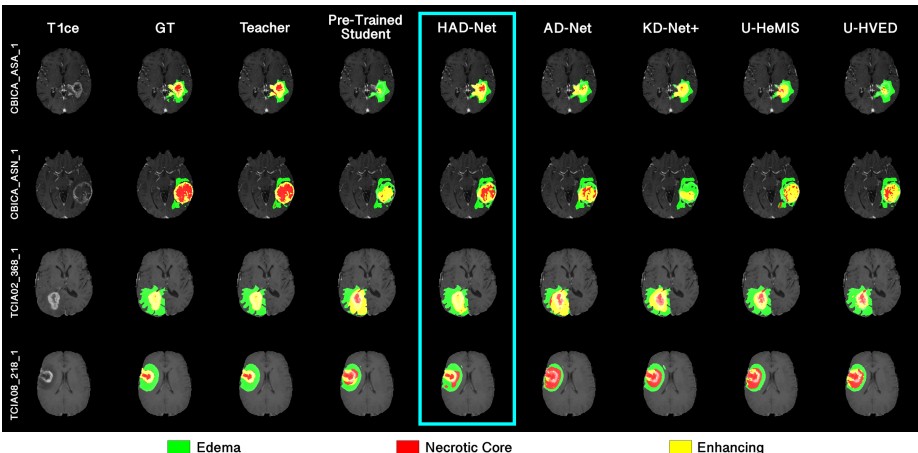

**Figure 2:** Qualitative comparison of different network outputs for MRI slices from 4 different cases belonging to the local testing set.

competing methods on a dataset where no manual segmentations are provided. Metrics for success are based on the segmentation of whole tumour (WT - ■ ■ ■), tumour core (TC - ■ ■), and enhancing tumour (ET - ■).

### 3.3. Quantitative Results

Table 1 shows the provided Dice coefficients for the proposed *HAD-Net* as well as the competing methods. As expected, the teacher, with access to all MRI sequences, has the highest Dice scores. However, *HAD-Net* outperforms all other competing methods in segmenting TC and ET. The overall effectiveness of knowledge distillation in this domain is depicted in its overall improvements over the pre-trained student network of 16.2% and 6.7% in Dice scores for ET and TC, respectively. *HAD-Net* also shows similar performance improvements over *AD-Net*, 18.7% for ET Dice and 3.3% for TC Dice, indicating the pivotal importance of the hierarchical component of the knowledge distillation. The proposed HAD-Net also shows statistically significant Dice performance gains over *KD-Net+*, *U-HeMIS*, and *U-HVED* by 17.2%, 18.9%, and 26.0%, respectively, for ET, and by 5.4%, 3.6%, and 8.4%, respectively, for TC. In fact, the paired t-tests between *HAD-Net*'s Dice scores and those of the other methods showed p-values of less than 0.001 for the ET Dice scores, and 0.05 for the TC Dice scores. This is evidence that *HAD-Net* is able to successfully distill meaningful information from the teacher to the student, ultimately allowing for better segmentation when contrast-enhanced images (T1ce) are unavailable. All methods performed similarly in terms of segmenting WT (no statistically significant difference), therefore *HAD-Net*'s gains in ET and TC segmentation performance were not attained at the expense of WT segmentation accuracy.

### 3.4. Qualitative Results

Although Dice scores provide a unified, objective measure for assessing the relative performances of different multi-class tumour segmentation models on the same dataset, they

|  | HAD-Net | AD-Net | Pre-Trained Student |
|---|---|---|---|
| Uncertainty Metric (↑) | 0.6084 * | 0.5894 | 0.5137 |

**Table 2:** Comparison of quantified uncertainty for enhancing tumour between HAD-Net, AD-Net, and pre-trained student output using the BraTS 2019 uncertainty quantification challenge metric (Mehta et al., 2020). (*) denotes statistically significant differences, where $p < 0.01$, between HAD-Net and all other methods, using a two-sided paired t-test.

do not convey the entire story. Qualitative comparisons of the results produced by various models permit examination of the subtle differences that can potentially have serious clinical implications. To that end, Figure 2 depicts the segmentation results produced by various models on 4 patient cases from the local testing set, where the slices of interest were chosen based on how clearly they depict the prominence of enhancing tumours. From these examples, *HAD-Net* clearly produces segmentation outputs that are most near to those of the teacher network and to ground truth (GT), as compared to other methods, particularly for enhancing tumour and necrotic core. In the example in the third row of Figure 2, in particular, only *HAD-Net* is able to correctly identify the absence of necrotic core, while all other competing methods incorrectly classify the center of the tumour as necrotic core. In the last row, one can see that the competing methods severely over-segment necrotic core, a mistake that is less pronounced for HAD-Net's output. That being said, this case does indicate that there is still room for improvement in the proposed model, as *HAD-Net* still confuses edema and necrotic core in some areas. This indicates that additional information remains to be learned from the teacher in order to avoid these types of errors.

### 3.5. Uncertainty Quantification

As expected, *HAD-Net* did not perform as well on enhancing tumour segmentation as the Teacher network, which has access to all MRI sequences. As such, it is important to quantify the uncertainties in the segmentation results to permit both downstream tasks and clinicians to assess the confidence of the system in the outputs. Therefore, it is essential that the uncertainties convey that when the system is confident in its assertions, it is correct, and that when it is not correct, it is less certain. Given this desired outcome, we now compare the quality of the uncertainties produced by *HAD-Net* with other competing models. To this end, we adopt the metrics used in the BraTS 2019 Uncertainty Quantification Challenge (Mehta et al., 2020).

Given that *HAD-Net*, *AD-Net*, and the pre-trained student network are trained using Dropout, Monte-Carlo-Dropout (Gal and Ghahramani, 2016) is used at test time to generate an uncertainty measure associated with their outputs[6]. The outputs are sampled 100 times, and entropies (Gal et al., 2017) are computed at each voxel. As the main drop in performance occurs in the segmentation of enhancing tumour when T1ce is not provided, focus is placed on the resulting uncertainties for this structure. To be consistent with the challenge, the uncertainties are normalized to lie between 0-100. Voxels are then filtered out

---

6. Please note that Dropout was not used in *U-HeMIS* nor for *U-HVED* and therefore they could not be used for comparison.

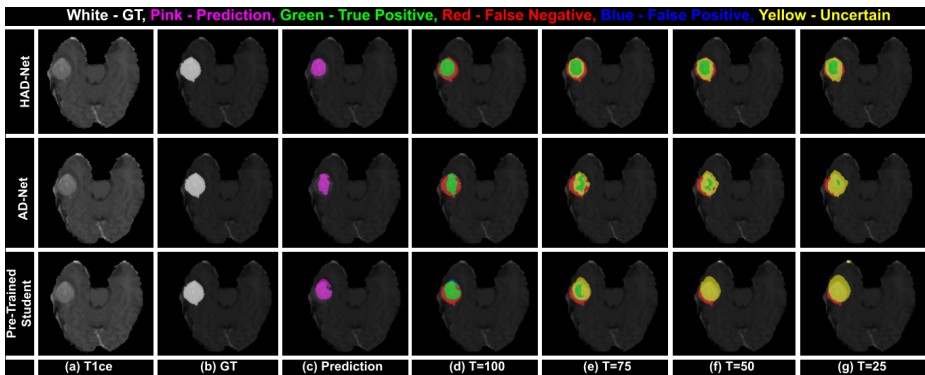

**Figure 3:** Comparisons of uncertainty thresholding on a example slice for enhancing tumour segmentation for *HAD-Net*, *AD-Net* and the pre-trained student: (a) T1ce MRI, (b) "Ground truth" label, (c) Prediction, (d) Prediction without filtering (T=100), and (e)-(g) Filtering with uncertainty thresholds (T) of 75, 50 and 25.

at different uncertainty thresholds (T=100, 75, 50, 25), such that all voxels with uncertainty values above T take on a new class value of *uncertain* and are removed from consideration in the segmentation metrics. The desired outcome is that once more uncertain voxels are filtered out, the segmentation performance on the remaining voxels should increase.

Table 2 shows the results comparing the uncertainty scores from the challenge for *HAD-Net*, *AD-Net* and the pre-trained student network. The scores range from 0-1, where a higher value is better (Mehta et al., 2020). *HAD-Net* shows statistically significant improvements over the other models. Figure 3 shows qualitative results on a case with one large enhanced tumour. As the uncertainty threshold is decreased (T=100, 75, 50, 25), *HAD-Net*'s more false negative voxels are marked as uncertain compared to true positive voxels. This leads to better performance on the remaining voxels. This is in contrast with *AD-Net* and the pre-trained student network, where, with the decrease in T, more true positive voxels are marked as uncertain compared to false negative voxels. This implies that *HAD-Net* is more confident in its correct assertions and more uncertain in its incorrect assertions.

## 4. Conclusions

In this paper, we introduced *HAD-Net*, a novel adversarial knowledge distillation network, where the teacher network teaches the student network, via hierarchical adversarial learning, how to better overcome the domain shift presented when key images are not available during inference. We applied the method to the open problem of brain tumour sub-tissue segmentation, where we showed significant performance improvements in segmenting enhancing tumours without T1ce over baseline methods, including recent modality-agnostic methods and knowledge distillation networks. The effect of the remaining errors was partially mitigated through uncertainty measures that reflected that the system is correct when confident and more uncertain when incorrect. The problem of segmenting enhancing tumours or lesions in medical images without contrast enhancing images is important in many clinical contexts, and there is room for further performance improvements. Future work will adapt HAD-Net to other cancers and neuro-degenerative diseases.

## Acknowledgments

This work was supported by a Canadian Natural Science and Engineering Research Council (NSERC) Collaborative Research and Development Grant (CRDPJ 505357 - 16), Synaptive Medical, and the Canada Institute for Advanced Research (CIFAR) Artificial Intelligence Chairs program.

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

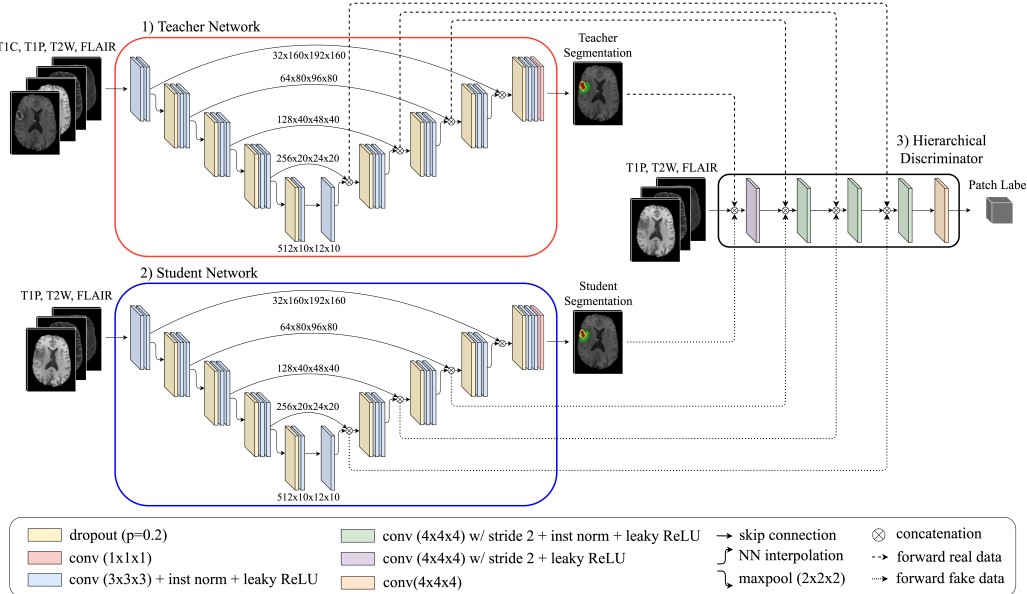

**Figure 4:** Diagram of the proposed HAD-Net. Note that components surrounded with a red border are frozen during training, and that only the components surrounded by a blue border are used during inference/testing.

## Appendix A. Network Architecture Details

As mentioned previously, the architectures of both the teacher network and the student network are adapted from the No New-Net model (Isensee et al., 2018). They each consist of four scales of encoder and decoder blocks, with a center block at the bottleneck of the U-Net (Çiçek et al., 2016) and an output block present after the final decoder block. The inner components of the encoder blocks, the decoder blocks, and the center block are identical. Each block, except the first encoder block, begins with a dropout layer (Srivastava et al., 2014) that has dropout probability p. This is followed by two CIL layers, where a CIL layer refers to the cascaded combination of a convolution layer, an instance norm layer (Ulyanov et al., 2016), and a leaky ReLU activation layer (Maas, Andrew L and Hannun, Awni Y and Ng, Andrew Y, 2013). In each block, convolution layers utilize a kernel size of 3x3x3 with $k * 2n$ filters, where n denotes the scale in which the block resides. The only difference between the three block types is the operations performed on their outputs. Encoder block outputs are passed through a maxpooling layer, with a kernel size of 2, prior to being forwarded to the next encoder block. On the other hand, decoder block and center block outputs are upsampled, via nearest neighbour interpolation, prior to being passed to the next decoder block. Finally, the aforementioned output block consists of a single convolutional layer with a kernel size of 1x1x1.

The HD consists of 4 discriminator blocks and a final output layer. The first discriminator block consists of a convolution layer and a leaky ReLU activation layer. The next 3 discriminator blocks consist of a convolution layer, an instance norm layer, and a leaky ReLU activation layer. In each of the discriminator blocks, the convolution layers utilize

a kernel size of 4x4x4 and a stride of 2, with $k * (2n + 1)$ filters. Lastly, the final output layer consists of a convolutional layer with a kernel size of 4x4x4 and a stride of 1. Unlike most discriminators, the HD does not assign a single label to classify the origin of the input segmentation and hierarchical latent features, instead it outputs a 3D patch label with a size relative to that of the input data. Notably, the HD does not utilize any pooling layers, rather, the stride lengths of the convolutional layers are used to downsample the outputs of a particular block to be half the size of its inputs. These dimensionality reductions ensure that the shape of the HD's feature maps match those of the segmentation networks at each scale. Ultimately, this allows for the hierarchical nature of the HD.

## Appendix B. Implementation Details

The pre-training procedure for the teacher and student networks are identical (with different input), with each training for a total of 400 epochs using a batch size of 1, using the same split as described in section 3.2. Both networks utilize data augmentation, where input modalities are randomly flipped and altered via random affine transformations. Furthermore, both networks have their initial number of filters, $k$, set to 32, their dropout probabilities, $p$, set to 0.2, and their leaky ReLU negative slopes set to 0.01. Moreover, they both utilize the same weighted CE loss functions, the same used in the $L_S$ loss function. It is important to note that the weights used for the CE loss are decayed by a factor of 0.98 after every epoch, until the weighted CE loss regresses to its unweighted form. Additionally, both networks use the same learning rate, which is initially set to a value of 0.0002. Also, both networks employ an AdamW (Loshchilov, Ilya and Hutter, Frank, 2017) optimizer with $\beta_1 = 0.9$, $\beta_2 = 0.999$, $\epsilon = 10^{-8}$, and a weight decay of $10^{-5}$. Finally, both networks utilize LR scheduling, where their respective learning rates are halved if they are unable to improve upon their best segmentation performance on the validation set, for 30 consecutive epochs.

Once pre-training is complete, the models that achieved the best performance on the local (held-out) validation set are selected to be used as the initial student and teacher networks in the training of HAD-Net. The training procedure for HAD-Net consists of training the student network and the HD for 800 epochs, using a batch size of 1. Unlike in the pre-training procedure, HAD-Net's training procedure does not utilize data augmentation, as empirically we found that training with data augmentation negatively affects the convergence of the student network. Therefore, we solely rely on dropout to provide sufficient regularization without adversely affecting convergence. Furthermore, HAD-Net does not utilize an LR scheduler to adjust the learning rate of the student network or the HD. Instead, both have a fixed learning of 0.0002. Adam (Kingma and Ba, 2014) optimizers are used for both the student network and the HD, with $\beta_1 = 0.5$, $\beta_2 = 0.999$, $\epsilon = 10^{-8}$, and a weight decay of 0. Note that these hyper-parameters are identical to those outlined in (Cirillo et al., 2020). For the student's loss function presented in Equation 1, we chose to use $\lambda = 0.2$, which we found to properly balance the CE and adversarial loss terms. In order to prevent the HD from becoming over-confident, the HD's parameters are not updated as often as the student. More specifically, for a given iteration, if the accuracy of the HD's labelling is above a certain threshold, the HD is not updated for that iteration (i.e., the $L_{HD}$ loss is not back-propagated).

## Appendix C.  Data Pre-Processing

The training, validation, and testing images were pre-processed prior to being fed to the segmentation models. For HAD-Net, AD-Net, KD-Net+, the pre-trained student, and the pre-trained teacher, the images were first center cropped to be of size 160x192x160. Then, for each MRI sequence, the mean and standard deviation of the brain region was computed. Next, each sequence was normalized by subtracting the mean and dividing by the standard deviation. Finally, the volume outside the brain region was set to 0. For U-HeMIS and U-HVED, the images were pre-processed according to the steps outlined in their respective papers (Havaei et al., 2016; Dorent et al., 2019).

## Appendix D.  Implementation Details for Competing Methods

In our work, we compare *HAD-Net* to several other methods: the teacher network (Figure 1(E)), the pre-trained student network (Figure 1(D)), *AD-Net* (Figure 1(B)), *KD-Net+* (Figure 1(C)), *U-HeMIS*, and *U-HVED*. Each of these methods were both trained (from scratch) and validated on the BraTS 2019 dataset, using exactly the same split as described in section 3.2.

- **Teacher:** The teacher network used for these comparisons is the same teacher used in HAD-Net to distill knowledge to the student network (i.e., the teacher network resulting from the pre-training procedure described in Appendix B).

- **Pre-trained student:** The pre-trained student network is simply the student network used at the very start of HAD-Net's training (i.e., the student network resulting from the pre-training procedure described in Appendix B).

- **AD-Net:** *AD-Net* is nearly identical to HAD-Net, with the only exception being that it utilizes a non-hierarchical discriminator, as opposed the hierarchical discriminator used by *HAD-Net* (i.e., it is a non-hierarchical version of *HAD-Net*). Consequently, *AD-Net*'s training procedure is exactly identical to that of *HAD-Net* (as described in Appendix B).

- **KD-Net+:** *KD-Net+* is a version of *KD-Net* (Hu et al., 2020) which we adapted to this context. More specifically, in *KD-Net+*, we apply the KD framework originally presented in the *KD-Net* paper to the student and teacher networks used in *HAD-Net*, by combining the loss functions and training procedure described in the original *KD-Net* paper with the student and teacher network architectures for *HAD-Net*. In order to train *KD-Net+*, we followed the training procedure detailed in the original paper (Hu et al., 2020); consequently, for this model, we used a frozen pre-trained teacher network and a randomly initialized student network.

- **U-HeMIS and U-HVED:** Finally, for our comparisons with *U-HeMIS* and *U-HVED*, we used the code made publicly available by the authors (GitHub Link). In order to be consisted with the proposed methods, we did not make any alterations to the code provided. These methods were trained (from scratch) and evaluated on the BraTS 2019 dataset but without additional hyper-parameter tuning.

