# OpenReview forum: "HAD-Net: A Hierarchical Adversarial Knowledge Distillation Network for Improved Enhanced Tumour Segmentation Without Post-Contrast Images"
_MIDL.io/2021/Conference — MIDL 2021_

### Official Review · AnonReviewer2 · 2021-03-06

**Confidence:** 4
**Preliminary Rating:** 3
**Recommendation:** Oral
**Final Rating:** 4

**Summary:**

This paper focuses on the problem of segmenting Enhancing Tumours (ET) from multimodal MRI images while not using the invasive T1ce (contrast enhanced) option as input. The proposed architecture uses teacher and student 3D UNets and a hierarchical discriminator that uses multiple levels of features from both the teacher and student networks. The idea is to use Knowledge Distillation to "teach" the student to produce good ET segmentations with limited input. The adversarial component of the architecture achieves that by means similar to GANs, where the student brings its segmentation close to the teacher segmentation due to the adversarial loss and the hierarchical discriminator. Note that the frozen teacher network needs to have had access to the T1ce input in its training.


**Strengths:**

Congratulations for your work. The paper is very well written. Related work is cited and used in validation, and the proposed architecture and results are very interesting, showcasing improved student network ET segmentation when compared to other methods that also are able to not use the T1ce input.

Figures are well made and follow a consistent style, clearly conveying qualitative results and the inner workings of the proposed architecture and related work.

The appendix with implementation and training details helps in keeping the main manuscript concise and focused on the novel aspects of this work, and the uncertainty task analysis brings more information supporting the potential of the HAD-Net architecture.


**Weaknesses:**

The authors claim all methods in validation were submitted to the BraTS validation system on the volumes without publicly available labels. However, the authors used a BraTS 2019 training set random split, while it seems that the architectures used for comparison were weights pre-trained in BraTS 2018 (if that’s not the case, the authors could make that more clear in the manuscript).

Another possible claim is that the T1ce input is being indirectly used in the HAD-Net, since the pre-trained teacher network uses T1ce inputs in its pre-training. In addition, those correspond to the same patients on the student’s training. In my opinion the HAD-Net methodology would be better validated using a pre-trained teacher network that was not trained by the authors in the exact same dataset as the student.


**Deanonymize Review:**

no

**Detailed Comments:**

A small change that might make the reading experience better for people that have not worked with the BraTS dataset before: the paper feels like it was written for the BraTS challenge submission, and does not explain very well that Enhancing Tumor and Necrotic Core are inside the Tumour Core (TC), or that the Whole Tumour label overlaps the TC and ET label. I understand that there might not be that much space for detailed explanations of the dataset, but maybe just putting Section 3.2 before the experiments, with a little more explanation of the data, would already help.

Something that might improve performance in general, and is used by some of the BraTS participants: some LGG cases have no positive ET groundtruth voxels. This can cause a 0 DICE value in the ET because of one false positive voxel, bringing the average DICE down. Some methods remove positive ET voxels when their volume is below a threshold. Might be worth having a look if you are having 0 ET DICE evaluations in some subjects by the BraTS submission system.

Finally, it is not clear if the student and teacher networks used the 160x192x160 crop as input, and if it is a center crop, or if there is use of patches. I assumed the whole crop is used due to the batch size of 1 and instance norm, but have seen works using smaller crops or patches when using Isensee’s 3D UNet.


**Final Rating Justification:**

The authors answered and clarified all my concerns related to this paper, which were the reasons i initially gave a weak accept.
Congratulations for your work.

**Justification Of The Preliminary Rating:**

Although there are some concerns and possible improvements, Knowledge Distillation is a hot topic with promising results in image processing in general. I believe the methodological idea showcased in this work, including the brilliant use of the discriminator and adversarial loss, is of interest to the medical image segmentation community.  Minor improvements and clarifications can improve the rating.


**Paper Type:**

both

**Questions To Address In The Rebuttal:**

Were the inputs to the student/teacher training center crops or patches?

Could the 2019 dataset be significantly better than the 2018 dataset, bringing an advantage to your method?

How do you address the argument that this method might be seen as actually using the T1ce, just in a complicated way (due to the fact the pre-trained network was trained in the same dataset as the student network)?

**Special Issue:**

yes

---

> ### Author Response · Authors · 2021-03-18
> **Reply to AnonReviewer2 (Part 1)**
>
> We would like to thank the reviewer for providing us with constructive feedback on our work.
>
> **Q. “Were the inputs to the student/teacher training center crops or patches?”**
>
> A. We use a center crop, such that we do not remove any of the brain from the image. We have updated the appendix to clarify this point.
>
>
> **Q. “Could the 2019 dataset be significantly better than the 2018 dataset, bringing an advantage to your method?”**
>
> A. We thank the reviewer for raising this point. While it is true that the baseline papers for KD-Net, U-HeMIS, and U-HVED show the methods being evaluated on the BraTS 2018 dataset, in this work, each of these models were both trained and evaluated on the BraTS 2019 dataset. We have further clarified this point  by adding a section to the appendix (Appendix-D).
>
>
> **Q. “How do you address the argument that this method might be seen as actually using the T1ce, just in a complicated way (due to the fact the pre-trained network was trained in the same dataset as the student network)?”**
>
> A. If we understood the question, indeed, the T1ce modality is available to the teacher during training. The student has to learn from the teacher how to perform better when T1ce is not available and therefore T1ce is available to the student indirectly during training. The important point is that, at test time,  inference is performed on a completely new dataset using only the student network which does not have access to the T1ce. Specifically, the model was evaluated on a completely new validation set at test time (BraTS 2019 Validation Set) which the student has never seen and for which we do not have access to the ground truth segmentations. We reworded the dataset section of the paper to further clarify this point. We hope this has answered the reviewer’s question.
>
>
> **Q. “A small change that might make the reading experience better for people that have not worked with the BraTS dataset before: the paper feels like it was written for the BraTS challenge submission, and does not explain very well that Enhancing Tumor and Necrotic Core are inside the Tumour Core (TC), or that the Whole Tumour label overlaps the TC and ET label. I understand that there might not be that much space for detailed explanations of the dataset, but maybe just putting Section 3.2 before the experiments, with a little more explanation of the data, would already help.”**
>
> A. We thank the reviewer for this comment. We have updated the paper to utilize color-coding to clarify this point (in section 3.2 of the paper).

---

> ### Author Response · Authors · 2021-03-18
> **Reply to AnonReviewer2 (Part 2)**
>
> **Q. “Something that might improve performance in general, and is used by some of the BraTS participants: some LGG cases have no positive ET groundtruth voxels. This can cause a 0 DICE value in the ET because of one false positive voxel, bringing the average DICE down. Some methods remove positive ET voxels when their volume is below a threshold. Might be worth having a look if you are having 0 ET DICE evaluations in some subjects by the BraTS submission system.”**
>
> A. We thank the reviewer for this suggestion. We have decided to construct a table that showcases the mean Dice scores achieved by each method on the BraTS 2019 Validation set, after post-processing (pp) their enhancing tumour segmentations by applying a threshold of 4000 voxels. The table is shown below. Note that only the ET DICE scores are shown, as the WT and TC DICE are unaffected by the post-processing. The first row of the table shows the ET DICE scores for the segmentation with post-processing, and the second row shows the ET DICE scores for the segmentation without post-processing. Note that the trend of HAD-Net outperforming all other competing methods is maintained and HAD-Net’s performance has indeed improved with post processing. We can include these results to the appendix (as well as a section explaining the post-processing procedure) at the reviewer’s request.
>
> |           | Teacher      | Pre-trained Student | HAD-Net            | AD-Net       | KD-Net+      | U-HeMIS      | U-HVED       |
> |-----------|--------------|---------------------|--------------------|--------------|--------------|--------------|--------------|
> | ET w/o pp | 70.1 +- 35.3 | 36.4 +- 27.7        | **43.6 +- 29.7** * | 35.3 +- 25.5 | 34.1 +- 25.9 | 34.3 +- 25.8 | 31.7 +- 23.4 |
> | WT w/ pp  | 73.3 +- 30.2 | 34.3 +- 25.3        | **39.8 +- 26.9** * | 33.5 +- 22.2 | 33.9 +- 22.9 | 33.5 +- 23.3 | 31.6 +- 20.9 |
>
>
> Table  showcasing  the  mean  Dice  scores  (%)  achieved  by  each  method  on  the BraTS 2019 Validation set (after post-processing on ET - threshold=4000) for whole tumour (WT), tumour core (TC), and enhancing tumour (ET). (*) denotes a statistically significant difference (p <0.001) between HAD-Net and all other methods, determined using a two-sided paired t-test.  Note that only the Teacher network had access to the T1ce sequence, and performed the best.

---

### Official Review · AnonReviewer3 · 2021-03-07

**Confidence:** 5
**Preliminary Rating:** 3
**Recommendation:** Poster

**Summary:**

Accurate segmentation of lesions requires usage of multiple modalities, in specific contrast injected modalities (T1 post contrast MRI). However, injecting contrast agents in patients is no longer thought to be safe. Hence, in this work, authors propose a network to provide competitive segmentation with all the modalities except the contrast injected one. To achieve this, authors have used a knowledge distillation framework, wherein the teacher trained on all the modalities transfers knowledge to the student trained on all the modalities but the post contrast injected MRI. The knowledge distillation is imparted through hierarchical discriminator by training it adversarialy. The authors have used the BraTS 2019 dataset for their experiments and they have compared their method with the recent developments in this line of research. Besides, they have also considered the uncertainty analysis.

**Strengths:**

* The paper is written well and the motivation is clearly presented.
* The authors have good knowledge on the literature.
* The authors have considered an interesting and an important problem in medical image analysis, providing better segmentation results without including contrast injected modality and this has a lot of potential.
* The usage of knowledge distillation (KD) for their problem is appreciable and their choice of hierarchical discriminator is an elegant one, as like the other KD techniques it doesn’t add any overhead in the test time.
* The methods considered for comparison are relevant ones and have covered the recent developments.
* The uncertainty analysis for the KD method is laudable as it indirectly provides information about the quality of distillation occured.


**Weaknesses:**

* As mentioned in the paper, adversarial KD techniques are widely present in literature. Bringing hierarchical discriminator into foray is interesting, but these kinds of discriminators are common in image segmentation techniques.
* As far as I understand, the current method requires three separate training, one for teacher, one for student and lastly training student with pre-trained weight through teacher and discriminator. I think, handling this would have added even more novelty to the work.
* The choice of fusing the feature maps from decoder blocks is understandable, but why only that feature maps, it can be any one of the following: 1) encoder, latent, and decoder, 2) encoder, decoder, 3) encoder, latent, 4) decoder, latent. I think some justification for this would be nice.


**Deanonymize Review:**

no

**Justification Of The Preliminary Rating:**

The authors have tackled an important problem in medical image analysis. The proposed method is appreciable and necessary experiments have been conducted, especially the uncertainty analysis. The method could have been more novel.

**Paper Type:**

methodological development

**Special Issue:**

no

---

> ### Author Response · Authors · 2021-03-18
> **Reply to AnonReviewer3**
>
> We would like to thank the reviewer for providing us with constructive feedback on our work.
>
> **Q. “As mentioned in the paper, adversarial KD techniques are widely present in literature. Bringing hierarchical discriminator into foray is interesting, but these kinds of discriminators are common in image segmentation techniques.”**
>
> A. We would like to point out that in the introduction, we bring attention to that fact that while hierarchical discriminators do exist, they are only found in the generative modeling literature (e.g. GANs for image synthesis, image segmentation) without knowledge distillation, and that while adversarial knowledge distillation techniques already exist, they are not hierarchical in nature. Consequently, to the best of our knowledge, HAD-Net is the first method to effectively combine these two components to permit KD to better overcome the large domain shift arising from the absence of crucial information during inference.
>
> In fact, by applying HAD-Net to the problem of tumour segmentation without contrast-enhanced images (i.e., T1ce), we show that the method is able to cope with the absence of key information significantly better than other KD methods. More specifically, in this context, where there exists a large gap between the student network and the teacher network, HAD-Net is able to significantly outperform a non-hierarchical version of our method (AD-Net), as well as the KD-Net framework (which utilizes a KL loss).
>
>
> **Q. “As far as I understand, the current method requires three separate training, one for teacher, one for student and lastly training student with pre-trained weight through teacher and discriminator. I think handling this would have added even more novelty to the work.”**
>
> A. We would like to thank the reviewer for this comment. We revised our paper to more clearly explain the implementation and training details (Appendix-B).
>
> Training an entire network end-to-end is common practice in traditional KD scenarios where the teacher and student network have different architectures but the same inputs. However, in this scenario, both the student and the teacher have the same architecture, but different inputs. Essentially, here, our goal is to bring the student network closer to the teacher network. By pre-training and subsequently freezing the teacher network, we ensure that it both provides adequate information to the student network throughout training and remains unaffected by the “pulling force” of the hierarchical discriminator (HD). If we were to train the network end to end, then the HD would not only pull the student closer to the teacher, but it would also pull the teacher towards the student. Ultimately, this would result in the two networks converging to a proverbial “middle-ground”. Empirically, we have found that doing this results in poorer performance from both the student and the teacher network. As for the reason behind pre-training the student, it simply provides a starting point for training HAD-Net.
>
> **Q. “The choice of fusing the feature maps from decoder blocks is understandable, but why only that feature maps, it can be any one of the following: 1) encoder, latent, and decoder, 2) encoder, decoder, 3) encoder, latent, 4) decoder, latent. I think some justification for this would be nice.”**
>
> A. The current architecture actually utilizes both the encoder and decoder latent representations. The latents of the decoder blocks are upscaled and concatenated with the latents of the encoder block prior to being forwarded to the hierarchical discriminator. While the architecture details are omitted from the diagram in Figure 1, a much more detailed diagram of the architecture is presented in the appendix (Figure 4).

---

### Official Review · AnonReviewer1 · 2021-03-08

**Confidence:** 4
**Preliminary Rating:** 3
**Recommendation:** Poster
**Final Rating:** 4

**Summary:**

This paper aims at solving the missing modality problem for brain tumor segmentation. In contrast to the synthesis-based methods, the proposed method is on the top of a teacher-student knowledge distillation architecture. It introduces a hierarchical multi-scale discriminator to enable a fine-grained distillation of latent information of teacher model towards the student model. Experimental results show it effectiveness w.r.t. segmentation performance and uncertainty estimation.

**Strengths:**

1.  The paper is well written and easy to follow. The methodological details and novelty are clarified clearly.
2.  The adopted hierarchical discriminator is technically sound. The multi-scale architecture is expected to provide an effective knowledge distillation from the teacher model to the student model.
3.  Experiments are performed extensively for segmentation performance and uncertainty quantification. Especially for uncertainty quantification, it would motivate the community to evaluate the reliability of the proposed method for downstream tasks, which brings noticeable benefits in clinical practice.


**Weaknesses:**

1.  From my point of view, the discriminator aims to minimize the distance (e.g., KL loss) between teacher and student model by adversarial learning. The similar strategy has been proposed in [1], and the major difference in this paper is the detailed implementation of discriminator, i.e. the hierarchical architecture, which seems to be relatively marginal for novelty.

[1] Chung, Inseop, et al. "Feature-map-level online adversarial knowledge distillation." International Conference on Machine Learning. PMLR, 2020.

2. Following the missing-modality setting, the goal is to improve the segmentation performance on enhancing tumor (ET). However, despite a considerable improvement over other methods, HAD-Net achieves only 39.81% Dice for ET, which seems to be unacceptable in clinical practice.
[1] Chung, Inseop, et al. "Feature-map-level online adversarial knowledge distillation." International Conference on Machine Learning. PMLR, 2020.



**Deanonymize Review:**

no

**Detailed Comments:**

1. The texts in Figure 1 are very small and hard to read.
2.  The proposed HAD-Net leads to a decrease of  segmentation accuracy for the whole tumor compared with KD-Net and AD-Net. Could the authors add some comments and discussions on this?

**Final Rating Justification:**

The contribution of using hierarchical discriminator for the knowledge distillation is clarified. The authors have also added experimental results that were not shown in the original submission.

**Justification Of The Preliminary Rating:**

The authors compared different variants of their method for experiments. The core idea is to use a hierarchical discriminator for the adversarial training problem, which was validated by experiments. A large improvement over its variants was observed.

**Paper Type:**

methodological development

**Questions To Address In The Rebuttal:**

1. Why not using a synthesis-based method for the segmentation task? The proposed method was not compared with such a method for dealing with missing modalities.
2. As mentioned above, the segmentation performance of ET was low in practice.

**Special Issue:**

no

---

> ### Author Response · Authors · 2021-03-18
> **Reply to AnonReviewer1 (Part 1)**
>
> We would like to thank the reviewer for providing us with constructive feedback on our work.
>
> **Q. “Why not using a synthesis-based method for the segmentation task? The proposed method was not compared with such a method for dealing with missing modalities.”**
>
> A. We thank the reviewer for this suggestion. We had originally removed comparisons with a synthesis-based model due to space limitations. Below, we show the results of an experiment comparing HAD-Net to a recent modality-synthesis model RS-Net [1].  RS-Net was used to synthesize T1ce MRI from T1,T2, and FLAIR sequences. For this experiment, we train a segmentation with all 4 real MRI and, during inference, the real T1ce image is replaced with the T1ce image synthesized by RS-Net. This permits assessing the performance of tumour segmentation with the synthetic T1ce image. Results can be found below. Note that HAD-Net shows both substantial and statistically significant improvements over RS-Net. This can be added to the appendix of the paper.
>
> [1] Raghav Mehta and Tal Arbel, “RS-Net: Regression-segmentation 3D CNN for synthesis of full resolution missing brain MRI in the presence of tumours.”, In International Workshop on Simulation and Synthesis in Medical Imaging, pages 119–129. Springer, 2018.
>
> |    | HAD-Net             | RS-Net       |
> |----|---------------------|--------------|
> | WT | 87.5 +- 07.6        | 86.0 +- 13.6 |
> | TC | **66.7 +- 19.7** *  | 57.5 +- 19.9 |
> | ET | **39.8 +- 29.6** ** | 28.7 +- 22.7 |
>
> (*) and (**) denote a statistically significant difference (p<0.05 and p<0.001) between the two methods, determined using a two-sided paired t-test. Note that the differences in WT DICE between the two networks is not statistically significant.
>
> **Q. “As mentioned above, the segmentation performance of ET was low in practice.”**
>
>
> A. Thank you for raising this concern. Please note that our goal was to present HAD-Net, not as a final solution to the problem of performing segmentation without contrast-enhanced images, but as a step toward finding this solution. To that end, the paper shows that using HAD-Net in this context yields significant advancements compared to the current state-of-the-art methods for ET segmentation (16% to 26%). We do agree, however, that further improvements will be required for full clinical deployment.
>
> We wish to stress that the problem presented in this paper has potential for broad clinical impact, given that contrast agents are used in many clinical domains (e.g. cancer, MS). However, the problem is very challenging and, as of yet, unsolved. Despite the fact that the student network does not yet perform as well as the teacher, we have shown that HAD-Net makes statistically significant improvements over other recently published methods on the same problem context. Although  this paper explores the context of brain tumour segmentation based on the relatively small BraTS dataset, the impact of the method on other domains is not yet known. As we will release the code, the method can be explored in other domains with potentially large implications. Furthermore, metrics such as Dice do not tell the whole story. We feel that automatic methods for segmenting pathological structures should properly reflect the confidence in the resulting segmentation prior to clinical deployment, in order to permit review by the clinical end-user. For this to be effective, the system should be correct when it is confident, and more uncertain when it is not. In this paper, we show that uncertainties produced by HAD-Net depict these traits. This should, at least in part, mitigate its performance drop over the teacher in that a clinician can assist in safely counting on the areas where HAD-Net is confident, and review the areas where it is not.

---

> ### Author Response · Authors · 2021-03-18
> **Reply to AnonReviewer1 (Part 2)**
>
> **Q. “From my point of view, the discriminator aims to minimize the distance (e.g., KL loss) between teacher and student model by adversarial learning. The similar strategy has been proposed in [1], and the major difference in this paper is the detailed implementation of discriminator, i.e. the hierarchical architecture, which seems to be relatively marginal for novelty.”**
>
> A. We would like to point out that in the introduction, we bring attention to that fact that while hierarchical discriminators do exist, they are only found in the generative modeling literature (e.g. GANs for image synthesis) without knowledge distillation, and that while adversarial knowledge distillation techniques already exist, they are not hierarchical in nature. We would like to clarify here that although the paper by Chung et al. [1] does present an adversarial KD technique, it does not utilize a hierarchical discriminator. Consequently, to the best of our knowledge, HAD-Net is the first method to effectively combine these two components to permit KD to better overcome the large domain shift arising from the absence of crucial information during inference.
>
> In fact, by applying HAD-Net to the problem of tumour segmentation without contrast-enhanced images (i.e., T1ce), we show that the method is able to cope with the absence of key information significantly better than other KD methods. More specifically, in this context, where there exists a large gap between the student network and the teacher network, HAD-Net is able to significantly outperform a non-hierarchical version of our method (AD-Net), as well as the KD-Net framework (which utilizes a KL loss).
>
> [1] Chung, Inseop, et al. "Feature-map-level online adversarial knowledge distillation." International Conference on Machine Learning. PMLR, 2020.
>
>
> **Q. “The proposed HAD-Net leads to a decrease of segmentation accuracy for the whole tumor compared with KD-Net and AD-Net. Could the authors add some comments and discussions on this?”**
>
> A. In this work, we focus on the challenging problem of improving enhancing tumour (ET) segmentation without the presence of T1ce, where we show that HAD-Net outperforms all competing methods in terms of TC dice and ET dice by margins that are both substantial and statistically significant. Although Table 1 shows that competing methods are comparable in terms of their segmentation performance on WT, all methods perform almost as well as the teacher in this case. Two methods perform marginally better than HAD-Net for WT segmentation, but the improvements are *not statistically significant* for this task. Perhaps the bold entry for the highest scores in Table 1 was misleading in this regard and the table was updated to more clearly make this point (i.e., only categories that have statistically significant differences will have the best model bolded).

---

### Official Review · AnonReviewer4 · 2021-03-09

**Confidence:** 4
**Preliminary Rating:** 2
**Final Rating:** 3

**Summary:**

The paper presents a segmentation model called HAD-Net which combines adversarial learning and knowledge distillation to segmentation brain tumors without post-contrast images. Like KD-Net, the model distills the learned features of a teacher trained with all modalities to a student trained with some missing modalities (T1CE in this paper). The main difference lies in the use of a hierarchical discriminator instead of a direct KL loss for enforcing feature similarity. The proposed method is tested on the BraTS 2019 Challenge dataset, for the task of tumor segmentation and uncertainty estimation.



**Strengths:**

* Interesting use of adversarial learning and knowledge distillation for segmentation with missing modality.

* Evaluation of the proposed method on two tasks: segmentation and uncertainty estimation. Results show statistically significant improvements for segmenting EC and TC, as well as uncertainty estimation.

* Paper is easy to follow.

**Weaknesses:**

* Technical contributions are somewhat limited. The proposed method is essentially KD-Net with the KL loss replaced by the hierarchical discriminator similar to (Valvano et al., 2020).

* The clinical motivation for this work is unclear. It is mentioned that injecting a contrast agent is unsafe for patients, however I would argue that having an inaccurate estimation of the tumor (e.g., in surgery planning) represents a greater risk. See next comment.

* Results are mixed. Accuracy is worse than baselines for WT. Although there is improvement for EC and TC, the Dice for these classes is really low (67% for TC, 40% for EC). Is this performance good enough for a real-life application?

* Important information about the baselines and their training is missing to properly assess the advantage of the proposed method.


**Deanonymize Review:**

no

**Detailed Comments:**

* Why not include the standard KD loss in the model? How can you be sure that the teacher and student will give consistent outputs?

* Since this the main methodological contribution of the paper, the use of a discriminator instead of a direct loss like l2 or KL divergence (as in KD-Net) should be better motivated ?

* Did you use the same pre-training strategy for KD? To better understand the improvements of the proposed method, it is necessary to provide additional details on this baseline as well as U-HeMIS and U-HVED (hyper-parameter tuning, backbone network, etc.)

* What is the impact of the lambda term? How was this hyperparameter chosen and is the method sensitive to its value?

* The experimental setting is different from the KD-Net paper, where the student has only T1ce. It is important to explain this difference since its impede direct comparison with previously published results.

* Since they are important to fully understand the method and reproduce results, I suggest moving Implementation details in the main paper.

* Why not use the latent features directly as input to D (possible upscaling them to have matching sizes) ? The added complexity of the proposed architecture should be better motivated.

* Table 1:  it would be better to include standard deviation and report HD95 accuracy (as in Brats challenge evaluation).

* Table 2: why not compare against KD-net? Also, I do not understand why Dropout cannot be added to U-HeMIS and U-HVED, so that these baselines can be included in the table.

**Final Rating Justification:**

I'm generally satisfied by the author's rebuttal and answers to my question. Although the paper is not perfect, I believe it is now suitable for acceptance.


**Justification Of The Preliminary Rating:**

The paper is interesting but there are several issues to address which include unclear clinical motivation, missing details on the implementation of baselines,  training setting different from the literature, etc.

**Paper Type:**

methodological development

**Questions To Address In The Rebuttal:**

* Better motivate of the proposed method given its low segmentation accuracy (compared to adding T1CE)

* If possible, compare the method in the same setting as the KD-Net paper.

* Add st. deviation an HD95 in Table 1

* If possible, compare the method against an architecture where the latent features of the Teacher and Student are made similar using a direct loss (e.g., KL or L2) instead of adversarial learning.

* Add information on the baseline implementation and training.

**Special Issue:**

no

---

> ### Author Response · Authors · 2021-03-18
> **Reply to AnonReviewer4 (Part 1)**
>
> We would like to thank the reviewer for providing us with constructive feedback on our work.
>
> **Q. “If possible, compare the method in the same setting as the KD-Net paper”**
>
> A. We  thank the reviewer for this question as it encouraged us to formulate a clearer nomenclature for the KD-Net model. The original KD-Net framework was devised for the (different) setting of knowledge distillation from multi-modal to mono-modal segmentation models. Our objective in this work was to compare KD-Net’s  knowledge distillation framework (loss functions and training procedure) to HAD-Net’s in the setting for this work (i.e., segmentation when missing T1ce). To this end, and for fair comparison, we adapted the KD-Net model by replacing its student and teacher networks with HAD-Net's while maintaining its knowledge distillation formulation. To make things clearer, we renamed our “KD-Net” baseline to “KD-Net+”, and have added some explanations to the main paper (mainly the 5th line of section 3.1). We also gave a detailed explanation of the KD-Net+ method in section D of the appendix. We hope that these changes address all questions regarding KD-Net.
>
>
> **Q. “Better motivate of the proposed method given its low segmentation accuracy (compared to adding T1CE)”**
>
> A. Our goal was to present HAD-Net, not as a final solution to the problem of performing segmentation without contrast-enhanced images, but as a step toward finding this solution. To that end, the paper shows that using HAD-Net in this context yields significant advancements compared to the current state-of-the-art methods for ET segmentation (16% to 26%). We do agree, however, that further improvements will be required for full clinical deployment.
>
> We wish to stress that the problem presented in this paper has potential for broad clinical impact, given that contrast agents are used in many clinical domains (e.g. cancer, MS). However, the problem is very challenging and, as of yet, unsolved. Despite the fact that the student network does not yet perform as well as the teacher, we have shown that HAD-Net makes statistically significant improvements over other recently published methods on the same problem context. Although  this paper explores the context of brain tumour segmentation based on the relatively small BraTS dataset, the impact of the method on other domains is not yet known. As we will release the code, the method can be explored in other domains with potentially large implications. Furthermore, metrics such as Dice do not tell the whole story. We feel that automatic methods for segmenting pathological structures should properly reflect the confidence in the resulting segmentation prior to clinical deployment, in order to permit review by the clinical end-user. For this to be effective, the system should be correct when it is confident, and more uncertain when it is not. In this paper, we show that uncertainties produced by HAD-Net depict these traits. This should, at least in part, mitigate its performance drop over the teacher in that a clinician can assist in safely counting on the areas where HAD-Net is confident, and review the areas where it is not.
>
>
> **Q. “Add st. deviation and HD95 in Table 1”**
>
> A. We would like to thank the reviewer for their recommendation of adding the standard deviation to the DICE scores in Table 1. We believe this is a great point; consequently, we have added the standard deviations to Table 1 and have updated the PDF accordingly.
>
> HD95 only looks at the distance to the surface of the structure, which, in this setting, is less important than the DICE metric which considers all the voxels within the structure. In the interest of space, and as others report similarly ([1, 2, 3]), we focus on reporting results on the DICE. Based on the request, however, we tabulated the results for HD95 and found huge variabilities and poor overall averages for all baseline methods using this metric (as expected), with no statistically significant differences between them. Due to space constraints, we can include these results in the appendix if requested.
>
> [1] KD-Net: https://hal.telecom-paris.fr/hal-02899529/document
>
> [2] SegAN: https://arxiv.org/pdf/1706.01805.pdf
>
> [3] U-HVED: https://arxiv.org/pdf/1907.11150.pdf

---

> ### Author Response · Authors · 2021-03-18
> **Reply to AnonReviewer4 (Part 2)**
>
> **Q. “If possible, compare the method against an architecture where the latent features of the Teacher and Student are made similar using a direct loss (e.g., KL or L2) instead of adversarial learning.”**
>
> A.  We are not entirely certain of what experiment the reviewer is requesting. If the reviewer is requesting experiments with a model that utilizes a direct loss (i.e., KL or L2) solely on the bottleneck of the student and teacher networks, we would like to point out that we have already shown that HAD-Net is able to outperform a model which does this (as this is what was proposed in KD-Net). However, assuming that the reviewer is requesting experiments with a model that utilizes a hierarchical KL loss (i.e., KL applied to each scale) instead of a hierarchical discriminator (HD), we ran a set of experiments for a new model, which we denote KL-Net. The results are presented below. In KL-Net, we utilize the same student and teacher architectures as HAD-Net, but instead of using the HD to distill knowledge, we use a combination of a KD-loss (the same one used in the KD-Net paper) and a KL-Loss applied at each scale of the student and teacher network hierarchy, rather than only at the bottleneck. The results are presented below. One can see that KL-Net is significantly less effective at knowledge distillation over  using the hierarchical discriminator. Note that this can be added to the appendix of the paper if requested.
>
> |    | HAD-Net             | KL-Net       |
> |----|---------------------|--------------|
> | WT | 87.5 +- 07.6        | 88.6 +- 09.5 |
> | TC | **66.7 +- 19.7** *  | 62.4 +- 21.1 |
> | ET | **39.8 +- 29.6** ** | 33.5 +- 23.8 |
>
> (*) and (**) denote a statistically significant difference (p<0.05 and p<0.001) between the two methods, determined using a two-sided paired t-test. Note that the differences in WT dice between the two networks is not statistically significant.
>
>
> **Q. “Add information on the baseline implementation and training.”**
>
> A.  We would like to thank the reviewer for this comment. We revised our paper to more clearly explain the implementation details pertaining to the competing methods, and added this information to the appendix (Appendix-D). Please note that all competing methods were both trained (from scratch) and validated on the BraTS 2019 dataset, using exactly the same split as described in section 3.2
>
> **Q. “Technical contributions are somewhat limited. The proposed method is essentially KD-Net with the KL loss replaced by the hierarchical discriminator similar to (Valvano et al., 2020).”**
>
> A. We would like to point out that in the introduction, we bring attention to that fact that while hierarchical discriminators do exist, they are only found in the generative modeling literature (e.g. GANs for image synthesis) without knowledge distillation, and that while adversarial knowledge distillation techniques already exist, they are not hierarchical in nature. We would like to clarify here that the paper by Valvano et al. [1] is a work that falls into the first of these two categories, as it does indeed utilize a hierarchical discriminator, but it uses it in a generative context (i.e., to generate segmentations), and not in the context of KD. Consequently, to the best of our knowledge HAD-Net is the first method to effectively combine these two components to permit KD to better overcome the large domain shift arising from the absence of crucial information during inference.
>
> In fact, by applying HAD-Net to the problem of tumour segmentation without contrast-enhanced images (i.e., T1ce), we show that the method is able to cope with the absence of key information significantly better than other KD methods. More specifically, in this context, where there exists a large gap between the student network and the teacher network, HAD-Net is able to significantly outperform a non-hierarchical version of our method (AD-Net), as well as the KD-Net framework (which utilizes a KL loss).
>
> [1] Gabriele Valvano, Andrea Leo, and Sotirios A Tsaftaris. Weakly supervised segmentation with multi-scale adversarial attention gates. arXiv preprint arXiv:2007.01152, 2020.

---

> ### Author Response · Authors · 2021-03-18
> **Reply to AnonReviewer4 (Part 3)**
>
> **Q. “Why not include the standard KD loss in the model? How can you be sure that the teacher and student will give consistent outputs?”**
>
> A. We assume the reviewer is referring to a standard KD loss, which is a distillation loss on the difference between the soft student predictions and the soft teacher labels. This is indeed an interesting suggestion; however, HAD-Net already includes KD for both the segmentations and the multi-scale latent features, including those from the highest level, as these are forwarded from the student and the teacher models to the hierarchical discriminator. As such, adding this additional loss explicitly on the output of the networks seems to be redundant. Nonetheless, we will try to explore this experiment explicitly in future work.
> We are not sure we follow the latter part of the comment regarding “consistent outputs”. Our student and teacher outputs are made consistent through the hierarchical discriminator. Please elaborate a little further on what you mean by “consistent outputs” in case our explanation does not answer your question.
>
> **Q. “Accuracy is worse than baselines for WT.”**
>
> A.  Although Table 1 shows that competing methods are comparable in terms of their segmentation performance on WT, all methods perform almost as well as the teacher in this case. Two methods perform marginally better than HAD-Net for WT segmentation, but the improvements are *not statistically significant* for this task. Perhaps the bold entry for the highest scores in Table 1 was misleading in this regard and the table was updated to more clearly make this point (i.e., only categories that have statistically significant differences will have the best model bolded).

---

### Meta-Review · Area_Chair1 · 2021-03-27

**Recommendation:** Accept (Poster)

**Metareview:**

This paper presents a good contribution in Transfer Learning and Domain Adaptation. It adresses Knowledge Distillation which is a hot topic  in image processing.  The proposed method is evaluated on two tasks: segmentation and uncertainty estimation.
The reviewers have noted that the paper is well written and the whole approach technically sound and relevant, but: technical contributions are somewhat limited, result are mixed, information on the baseline validation and  justification of some methodological choice are missing.

The authors have clarified their paper with explanations and additional experiments, that the reviewers have  acknowledged.

 I recommend acceptance of this paper.


**Paper Type:**

both

---

### Decision · Program_Chairs · 2021-03-31

Accept